# Defining Fluoroquinolone Resistance-Mediating Mutations from Non-Resistance Polymorphisms in *Mycoplasma hominis* Topoisomerases

**DOI:** 10.3390/antibiotics10111379

**Published:** 2021-11-10

**Authors:** Martin Sharratt, Kirsty Sands, Edward A. R. Portal, Ian Boostrom, Brian A. Mondeja, Nadia M. Rodríguez, Lucy C. Jones, Owen B. Spiller

**Affiliations:** 1Division of Infection and Immunity, School of Medicine, Cardiff University, Cardiff CF14 4XN, UK; SharrattMG@cardiff.ac.uk (M.S.); kirsty.sands@zoo.ox.ac.uk (K.S.); PortalE@cardiff.ac.uk (E.A.R.P.); boostrom@cardiff.ac.uk (I.B.); 2Department of Zoology, Oxford University, Oxford OX1 3RE, UK; 3Bacteriology Reference Department, UK Health Security Agency, London NW9 5EQ, UK; 4Pedro Kourí Tropical Medicine Institute, Havana 11400, Cuba; brian.mr@cea.cu (B.A.M.); nrodriguezpreval@gmail.com (N.M.R.); 5Center for Advance Research of Cuba, Havana CP17100, Cuba; 6Department of Integrated Sexual Health, Cwm Taf Morgannwg University Health Board, Pontypridd CF37 1LB, UK; lucycjones@doctors.org.uk

**Keywords:** *Mycoplasma hominis*, epidemiology, United Kingdom, genomics, antibiotic resistance, genome analysis, topoisomerase, fluoroquinolone

## Abstract

Often dismissed as a commensal, *Mycoplasma hominis* is an increasingly prominent target of research due to its role in septic arthritis and organ transplant failure in immunosuppressed patients, particularly lung transplantation. As a mollicute, its highly reductive genome and structure render it refractile to most forms of treatment and growing levels of resistance to the few sources of treatment left, such as fluoroquinolones. We examined antimicrobial susceptibility (AST) to fluoroquinolones on 72 isolates and observed resistance in three (4.1%), with corresponding mutations in the quinolone resistance-determining region (QRDR) of S83L or E87G in *gyrA* and S81I or E85V in *parC*. However, there were high levels of polymorphism identified between all isolates outside of the QRDR, indicating caution for a genomics-led approach for resistance screening, particularly as we observed a further two quinolone-susceptible isolates solely containing *gyrA* mutation S83L. However, both isolates spontaneously developed a second spontaneous E85K *parC* mutation and resistance following prolonged incubation in 4 mg/L levofloxacin for an extra 24–48 h. Continued AST surveillance and investigation is required to understand how *gyrA* QRDR mutations predispose *M. hominis* to rapid spontaneous mutation and fluoroquinolone resistance, absent from other susceptible isolates. The unusually high prevalence of polymorphisms in *M. hominis* also warrants increased genomics’ surveillance.

## 1. Introduction

*Mycoplasma hominis* is an uncommon urogenital colonizer that belongs to the *Mollicutes* class of bacteria. While pathogenicity in immunocompetent individuals is controversial, it has recently been linked to bacterial vaginosis [1]. Infection can also increase the risk of female infertility, spontaneous abortion, stillbirth, and premature rupture of membranes [2]. While viewed as a pathobiont (i.e., organisms that can cause harm under certain circumstances), the association of *M. hominis* with failure in lung transplant patients or potentially lethal hyperammonemia means that it is important to develop fast and effective methods of treatment for the disease in immunosuppressed patients [3,4,5].

Mollicutes are one of the simplest forms of self-replicating life, and as such are resistant to most of the treatment methods normally employed by clinicians. They lack the cell walls that are targeted by beta-lactams and glycopeptides and lack the folic acid pathways that would be inhibited by sulphonamides and trimethoprim [6]. *M. hominis* itself is naturally resistant to 14- and 15-membered ring macrolides [7] and incredibly fastidious, making in vitro growth and/or detection incredibly laborious. The remaining therapeutics for treatment are macrolides, tetracyclines, and fluoroquinolones, the last of which is the family to which ciprofloxacin, levofloxacin, and moxifloxacin belong.

Fluoroquinolones target the type II topoisomerases that facilitate alterations in chromosomal supercoiling necessary for transcription and DNA replication. By binding to these topoisomerases, they render the topoisomerases unable to disassociate from the DNA molecule so it cannot reform, creating wide-scale nucleotide breaks and ultimately cell death. While effective therapeutics for *M. hominis*, they can be associated with a variety of adverse side effects. These include tenonitis, tendon rupture, potential prolonged QT intervals, and, rarely, cardiac arrhythmia [8]. This means that application of this treatment needs to be highly targeted and account for issues such as drug resistance.

The mechanism for fluoroquinolone resistance in *M. hominis* has been established as arising from Single Nucleotide Polymorphisms (SNPs) in the Quinolone Resistance-Determining Region (QRDR) of the topoisomerase genes, as best characterized for *E. coli* but also reported for *M. hominis* [9,10,11]. By altering the protein conformation of the resulting topoisomerases, they become immune to the disruptive effect of fluoroquinolones. As this mutation can arise with a single-point mutation, it can occur spontaneously in clinical isolates regardless of lineage, and, therefore, rapid analysis of isolates during treatment is the most reliable way of identifying these resistant strains. 

To aid in this analysis, we analyzed the entire *gyrA*, *gyrB*, *parC*, and *parE* genes (including the QRDR) of 72 different sequenced strains and found mutations in the QRDR of three strains that conferred phenotypic antibiotic resistance to fluoroquinolones. We further showed evidence of fluoroquinolone-susceptible strains carrying QRDR mutations in the *gyrA* gene that were able to spontaneously gain a *parC* mutation after prolonged incubation with higher concentrations of levofloxacin, which was not observed in isolates without *gyrA* mutations. We also investigated the prevalence of non-resistance polymorphisms in these key genes for *M. hominis*, which could obfuscate a genomics-based approach to resistance screening for this bacterium. 

## 2. Results

Overall, of the 72 *M. hominis* isolates that were analyzed and came from a wide range of geographic locations 12 were isolated from Havana, Cuba; seven from Pancevo, Serbia; 31 from various locations in England, five from Perth, Australia; and 14 from Pontypridd, Wales (as well as the ATCC prototype reference strains from France and the USA). Many were collected as part of previous studies and the metadata are included in the Appendix A. The full list of non-synonymous polymorphisms present for *gyrA*, *gyrB*, *parC*, and *parE* in all 72 isolates, as compared to ATCC 23114, are also given in Appendix B, Table A1, Table A2, Table A3, Table A4, Table A5 and Table A6. Reference strain ATCC 33131 (Sprott Strain) was also examined as a prototype strain. Table 1 lists the variations in SNP frequency between all isolates, both for all SNPs present and for just functional/non-synonymous SNPs. Of the four genes analyzed, *gyrA* contained the highest number of SNPs of any of the four genes, with between 25 to 97 SNPs present in any one isolate relative to ATCC 23114 (Table 1). When we looked at non-synonymous SNPs (i.e., only those affecting changes in amino acid composition), *parC* contained the highest levels of variation with five to 10 amino acid polymorphisms in any one isolate relative to ATCC 23114 (Table 1).

Of the 72 isolates that were initially analyzed, three demonstrated phenotypic resistance to fluroquinolones (levofloxacin MIC > 2 mg/L and moxifloxacin MIC > 0.5 mg/L): U006, MH10-09, and MH15-03. All other isolates demonstrated conventional susceptibility to fluoroquinolones, and these three isolates were found to have non-synonymous mutations in both the *gyrA* and *parC* genes (Figure 1). However, full gene analysis also identified two isolates, DF28 and S019M, that carried the same *gyrA* mutation observed in U006 and MH15-3 (S153L; *E. coli* numbering S83L). They had no significant elevation in levofloxacin MIC of 0.5 mg/L (relative to mean of 0.352 ± 0.17 mg/L for all susceptible isolates; Figure 2), although they both had an intermediate moxifloxacin MIC of 0.25 mg/L (relative to mean of 0.80 ± 0.01 mg/L for all susceptible isolates, Figure 2). 

Furthermore, prolonged incubation of these two isolates with levofloxacin resulted in spontaneous induction of resistance (MIC 4 or 8 mg/L for moxifloxacin and levofloxacin, respectively; Figure 2) within 24–48 h later, which was not observed for other isolates. Resequencing of these induced resistant isolates (given names, DF282R and S019M2R) and found induction of the same *parC* mutation as observed for MH10-9 S91I (S81I *E. coli* numbering; Figure 1). While the *gyrA* SNP of MH10-09 was observed to be shifted 12 bases downstream relative to the QRDR mutations present in other isolates, no significant difference in MICs were detected between these three isolates. It is interesting to note that while S019M and DF28 shared the observed *gyrA* mutation with MH15-3 and U006, the induced *parC* mutation aligned with that observed for MH10-9 (Appendix B, Table A1, Table A3, Table A4 and Table A5). Further, U006 additionally contained an K144R mutation proposed as a source of fluoroquinolone resistance in a previous report [12], but this polymorphism can be observed in 28 other isolates (Appendix B, Table A3, Table A4 and Table A5) without elevated MICs in this cohort and, therefore, is clearly a non-resistance polymorphism and not a resistance-mediating mutation. With regard to *parC* QRDR mutations, this appears to be the dominant determinant for levofloxacin and moxifloxacin resistance as only those strains carrying either S91I (*E. coli* numbering S81I) or E95K or V (*E. coli* numbering E85K or V) were phenotypically resistant. The *gyrA* mutation S153L is only capable of mediating an intermediate moxifloxacin MIC = 0.25 mg/L when present alone (Figure 2). 

As demonstrated by Figure 3, phylogenetic variance between resistant isolates varied significantly. DF28 and DF282R did not cluster together along the same branch as S019M and S019M2R even though both pairs of isolates were a product of spontaneous resistance induction. The naturally occurring resistant isolates (MH10-09, MH15-03 and U006) were spread throughout the phylogenetic tree, indicating a low chance of resistance being shared along a subgroup or to represent clonal expansion.

## 3. Discussion

Single mutation in the *GyrA* QRDR was found associated with susceptible fluoroquinolone MICs and resistance was found to require an additional *ParC* QRDR mutation. Unfortunately, no isolates with an isolated *ParC* QRDR mutation were identified to examine. Furthermore, only isolates with the pre-existing *GyrA* QRDR mutation spontaneously developed induced resistance in our study. We examined resistance against levofloxacin and moxifloxacin, despite the fact these are rarely used to treat sexually transmitted infections (ciprofloxacin or ofloxacin being more frequently prescribed). This highlights a short fall of the only available internationally agreed thresholds for fluoroquinolone resistance determination [13]. However, MICs for ciprofloxacin are usually the same or slightly higher than those observed for levofloxacin. In our study, of the 72 isolates that underwent AMR testing, only 4% of isolates were identified as being resistant to fluoroquinolones. While this appears much lower than in some other studies that specifically investigated fluoroquinolone resistance rates [12,14,15], our rates are consistent with those observed in a recent multi-national study on patients undergoing infertility investigation and symptomatic sexual health patients in the UK, France, and Serbia in 2019 (resistance in 2/85 *M. hominis* isolates) [16]. Furthermore, a study examining 1000 sexual health patients in Wales identified 100 *M. hominis* isolates with no fluoroquinolone resistance [17]. Our study contained samples from a variety of geographical regions, which included all previously available, archived fluoroquinolone-resistant isolates, but were chosen because they had associated whole genome sequences, and so is not representative of a complete clinically relevant data set. Hopefully more in depth sequencing studies for laboratories reporting higher fluoroquinolone resistance rates will be forthcoming.

Of the three resistant isolates and two inducible isolates, there was a consistent modification of S153L of the QRDR section of *gyrA*, with MH10-09 being the exception having a E157G mutation. While not entirely uniform, the narrow band of mutation positions provides a starting point for further analysis regarding the functional effects of a mutation in this region. Mutations in the *parC* genes of these isolates were more varied, with S91I mutations observed in U006 and MH15-03, but a E95V mutation observed in MH10-09. This is further supported by the observation of variable resistance observed in S019M and DF28, which contained the exact same set of SNPs in the QRDR section of its *gyrA* region but had an E95K mutation, rather than an E95V mutation, in its *parC* gene. The earliest investigation of induced fluoroquinolone resistance in *M. hominis* through multi-step repeated challenge of isolates with ciprofloxacin, norfloxacin, pefloxacin, and ofloxacin, by Bebear et al., only reported *gyrA* QRDR mutations [18]. However, this ground-breaking study in 1999 did not have the ability to additionally amplify and examine the *parC* gene for the same resistant isolates to see if corresponding mutations were present for that gene as well. These historical studies relied on Sanger sequencing of amplicons using primers designed against conserved regions from other bacteria, and our study has the benefit of long sequence contigs from high-depth next generation whole genome sequencing to identify the full range of polymorphisms across all gyrase and topoisomerase genes. 

Similar examples of co-mutations contributing to a variance of antimicrobial resistances have been observed in previous studies of *M. hominis*. Observations specifically regarding the gyrase genes in other human mollicutes have been made before [14,19,20], but the variance of resistance and its corresponding regulation is, to our knowledge, a novel observation we have made here. More broadly, other mutations have been implicated as the source for fluoroquinolone resistance such as *parC* K144R mutation [12], which we found in resistant strain U006, but also in 28 susceptible strains. These authors also reported A154T *parC* mutation as a potential resistance marker; while we did not find any A154T mutations, we did find six susceptible strains carrying the A154V polymorphism. We identified many other polymorphisms but did not find the putative *parE* mutation A463S in any susceptible or resistant isolate.

When we look more broadly at the comparisons between isolates, one observation we can make is the significant heterogeneity in the reference genomes that are conventionally used for *Mycoplasma hominis.* Of the 72 isolates used, two reference strains were included, ATCC 23114 and Sprott. Although reference strains, both exhibited incredibly high levels of heterogeneity relative to each other, which further exemplifies the unusually high polymorphism prevalence of this species. More broadly, QRDR-mediated resistance is a common occurrence in a variety of similar pathogens and similar amino acid changes are conserved across multiple species (which is the basis for determining the *E. coli* numbering equivalents). QRDR-mediated mutations have been observed in *E. coli* [9] as well as *Streptococcus* spp. [21] and *Enterococcus faecalis* [22], each of which has been suggested as a putative source of the original gyrase/topoisomerase complex via non-specific horizontal gene transfer [23,24]. The seemingly non-specific nature of horizontal gene transfer implies there may be a relationship between antimicrobial resistance development and co-infections. However, this mechanism was not supported by our study despite the high inter-isolate variability, as the topoisomerase and gyrase genes are still much more closely related to each other than to out-groups by phylogeny analysis (Figure 3).

As demonstrated in Figure 3, the genetic diversity of *M. hominis* is incredibly high, more so than one would expect. The unusually high diversity for *M. hominis* has been noted before. Multi-locus sequence typing schemes developed for *M. hominis* have previously reported that identical sequence types are only found when isolated from the same patients at multiple timepoints [25]. We found the highest degree of variation in the *gyrA* gene (the lowest having 25 and the highest having 97 SNPs relative to ATCC 23144), which is in sharp contrast to the *gyrA* gene for the closely related urogenital mycoplasma *Ureaplasma* spp. [20]. Comparative investigation of the *gyrA* gene for 51 *Ureaplasma parvum*, for which we had whole genome sequences, found a range of only 0–7 SNPs between all isolates (data not shown). What is notable from a phylogenetic standpoint is the distinct lack of interrelation between the resistant isolates, beyond the interrelatedness of induced resistance-linked isolates. This not only reiterates the highly variable nature of *M. hominis* as an organism, but also the variable nature of the resistance-determining polymorphisms we observe in resistant isolates. Antimicrobial resistance in *M. hominis* conferred via in vitro selection pressure is well documented [18]. However, the finding that pleiotropy observed between *gyrA* and *parC* does not seem to correlate with any sort of genomic interrelatedness means that the ability of *M. hominis* to develop fluoroquinolone resistance so rapidly could itself be considered a mechanism of resistance in tandem with the QRDR polymorphisms themselves.

The ability for previously susceptible variants of *M. hominis* to develop clinically relevant levels of resistance to fluoroquinolones demonstrates the need to expand our surveillance methodologies to include the capacity for identifying these potentially resistant variants before treatment is undertaken, to prevent resistance developing against one of the few therapeutics we have left to combat this pathogen.

## 4. Materials and Methods

With the exception of the *M. hominis* strains isolated in Havana, Cuba, the remaining strains were archived from previously published studies [16,17,26,27]. Recovery of frozen archived isolates was performed via resuspension in Mycoplasma Experience Limited (Reigate, UK) selective media. Plates sealed with clear adhesive film were incubated in a humidified chamber at 37 °C for up to 5 days. Cultures and plates were checked daily, with their growth recorded. Growth in broth culture was visualized as a yellow to red color change in the absence of turbidity.

Antimicrobial screening was performed as outlined by CLSI guidelines [13] as per previous publication [27,28]. Mycoplasma selective medium used for MIC determination via broth microdilution method was provided by CPM SAS (Rome, Italy). In total, 72 archived *M. hominis* strains were analyzed. Induction of fluoroquinolone resistance was performed similarly to the agar-based, single-step induction method described by Bebear et al. [18], except that our selection occurred in microbroth dilution conditions. CLSI guidelines require determination of susceptibility at a strict 24–48 h incubation point. However, by allowing incubation of the MIC plates a further 24–72 h beyond MIC determination, spontaneous color change was observed for wells of 4 mg/L for levofloxacin. Sub-culturing these isolates found they now had an MIC of 8 mg/L. However, this phenomenon was only observed for isolates that already had *gyrA* QRDR mutations.

Strains to be examined by whole-genome sequencing were grown in 30 mL of Mycoplasma selective medium, pelleted at 13,000× *g* for 3 h, and resuspended in 400 µL of sterile distilled water as the first step of DNA extraction using the Qiagen EZ1 Advanced XL automated extractor utilizing the EZ1 DSP virus kit, as per the manufacturer’s instructions. DNA yields were between 1 and 8 ng/mL (Qubit 4.0; ThermoFisher, Loughborough, UK). Genomic libraries were pre*parE*d using Nextera XT v2 (Illumina, San Diego, CA, USA), with a bead-based normalization, following manufacturer guidelines. Paired-end WGS was performed on an Illumina MiSeq using the v3 chemistry to generate fragment lengths up to 300 bp (600 cycles). The bioinformatics’ pipeline used FastQC v0.11.8 and Trimgalore v0.5.0 [29,30] to validate and trim the raw sequence reads. The whole genome assembly and mapping pipeline consisted of Flash v1.2.11, SPAdes v3.12, BWA v0.7.17, pilon v1.23, and quast v5.0.0 [31,32,33]. Whole genome annotation and profiling of the genetic determinants of both fastq- and de novo-assembled reads was performed using prokka v1.14.0, NCBI BLAST, kmerfinder, CARD, srst2 v0.2.0, ARG-ANNOT, and VFDB [34]. The identification, annotation, and labelling of the QRDR and all SNPs (aligned and identified relative to ATCC 23114) was conducted using Geneious Prime software (Biomatters ltd. Auckland, New Zealand). The complete sequences of the *gyrA*, *gyrB*, *parC* and *parE* genes for all isolates examined are available in the Appendix A.

## Figures and Tables

**Figure 1 antibiotics-10-01379-f001:**
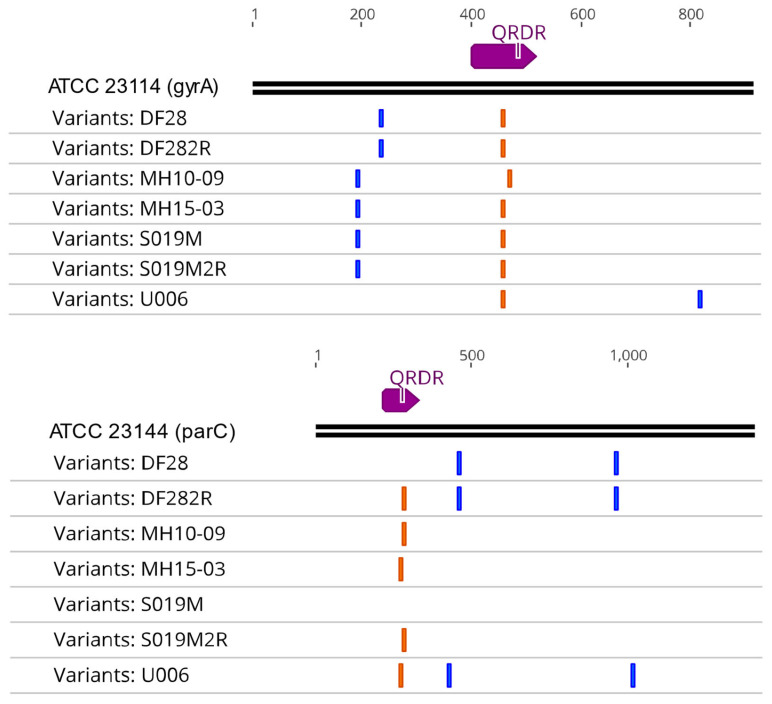
The single nucleotide polymorphisms (SNPs) of *gyrA* and *parC* genes found in the QRDR regions of all strains analyzed. Orange lines represent functional mutations, while blue lines represent synonymous mutations. No functional mutations were found in the QRDR regions of *gyrB* or *parE*.

**Figure 2 antibiotics-10-01379-f002:**
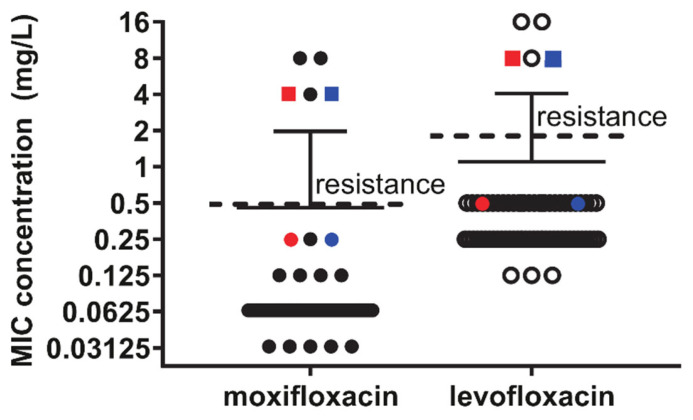
Minimum inhibitory concentration for *M. hominis* isolates for levofloxacin and moxifloxacin. Internationally agreed thresholds for resistance are shown as dotted lines and isolates above the line are resistant to the respective fluoroquinolones. Isolates with QRDR *gyrA*-only mutations are shown (S019M as red circles and DF28 as blue circles), and MICs for these isolates following one step induction of resistance, resulting in an additional QRDR *parC* mutation, are shown as colored squares (S019M2R as red squares and DF282R as blue squares).

**Figure 3 antibiotics-10-01379-f003:**
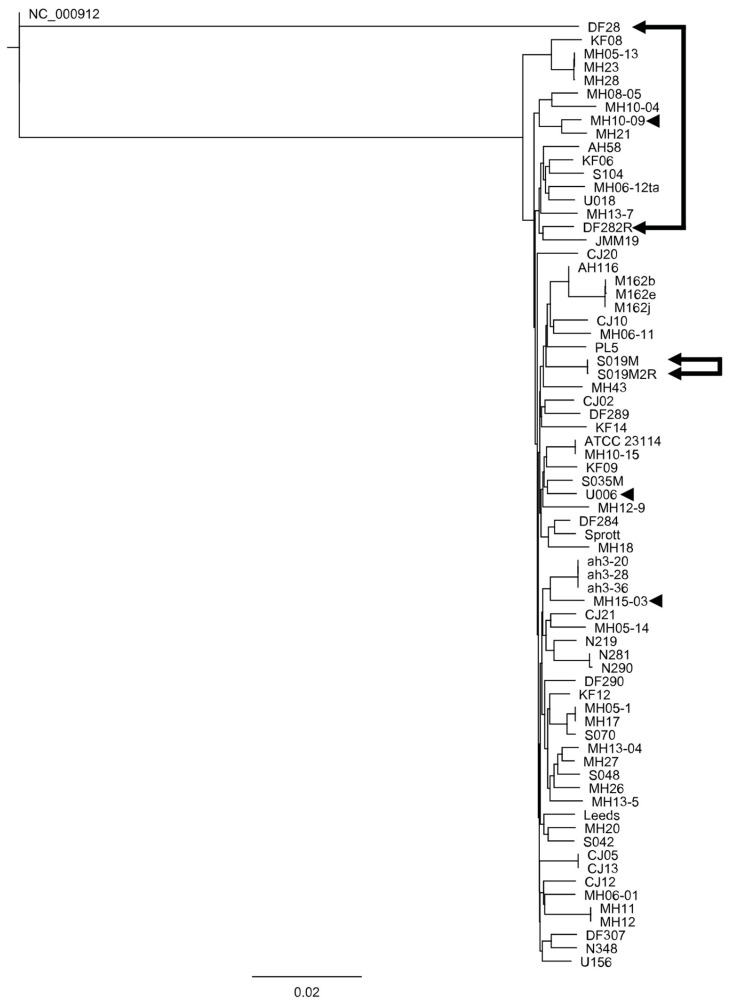
A phylogenetic tree of 72 isolates analyzed (and two induced resistant strains linked by arrows), constructed via concatenations of all four topoisomerase genes (*gyrA*, *gyrB*, *parC* and *parE*). The outgroup used was Mycoplasma pneumoniae M129, listed on the tree under its accession number (NC000912). Naturally occurring resistant strains are identified by arrowheads.

**Table 1 antibiotics-10-01379-t001:** A summary of the ranges of SNP frequencies observed across the topoisomerase genes of each of the 72 isolates analyzed.

Gene	Total SNPs	Non-Synonymous SNPs
*gyrA*	25–97	2–6
*gyrB*	1–34	0–3
*parC*	3–63	5–10
*parE*	14–62	1–6

## Data Availability

The data presented in this study, beyond data deposited as indicated in Genbank via accession numbers, are available on request from the corresponding author. The data that are not publicly available relate to patient confidentiality.

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
