# Peer review of "Defining Fluoroquinolone Resistance-Mediating Mutations from Non-Resistance Polymorphisms in Mycoplasma hominis Topoisomerases"

_antibiotics, 2021, doi:10.3390/antibiotics10111379_

Round 1
Reviewer 1 Report
The paper presents an analysis of the presence of single nucleotide polymorphisms (SNPs) in specific genes (gyrA, gyrB, parC and parE) of 72 isolates of Mycoplasma hominis and stablishes their connection with the observation of resistance to fluoroquinolones on 3 of these isolates. Additionally, the authors found 2 isolates that, after being exposed to fluoroquinolones, developed resistance to these drugs and spontaneously acquired a new SNP on the parC gene (in addition to a previous mutation on gyrA). The data showed in this paper allows to conclude that a combination of gyrA and parC mutations is required for Mycoplasma hominis to display resistance to fluoroquinolones. The paper contains interesting data and is fairly well explained but some details need to be further discussed/analysed.
- In the abstract the authors say: “However, there were high levels of polymorphism identified between all isolates outside of the QRDR, indicating caution for a genomics-led approach for resistance screening”. Why this never discussed in the manuscript?
- On page 2, line 62, the authors claim that the mechanism of resistance to fluoroquinolones in M. hominis has been established as arising from SNPs in the QRD region of the topoisomerase genes but they do not present any reference for this. The only reference they present is a review that briefly explains this mechanism of resistance for E. coli. The authors should change this sentence. This mechanism of resistance was characterized for E.coli and the authors assume that it may be important in M. hominis as well, but it is not correct to say that this is the established mechanism of resistance in M. hominis, if no reference is presented. Alternatively, the authors should cite the work that presents this mechanism has the stablished one in M. Hominis.
- Three isolates out of the 72 tested were resistant to levofloxacin and moxifloxacin, what about other fluoroquinolones, especially ciprofloxacin, were they tested? Why is ciprofloxacin later used to “trigger” resistance (according to the abstract) and it was not tested here? It would be important to have a table with the MICs (for all the fluoroquinolones tested) of the reference strains and, at least, these 3 isolates that showed resistance plus the 2 that carried the same gyrA mutation and developed resistance. I also suggest adding a supplementary table with the MICs for all isolates tested as it helps a lot following the results section.
- In the discussion the authors say, correctly, that ciprofloxacin and ofloxacin are the fluoroquinolones more relevant in this study as they are the most frequently used. Then, they say that they found MICs for ciprofloxacin of 16-32 mg/L but do not state which isolates were resistant to this fluoroquinolone, if they had any mutation of the QRD region and do not show the data. This data is important and should be integrated and discussed in the paper. Again, it is not clear at all why ciprofloxacin was not used as a “test” compound but it was used in the “resistance development” experiments.
- In the discussion the author acknowledge that they found a very low rate of resistance to fluoroquinolones in the set of samples they studied, when compared to other studies clinical isolates and that this is remarkable but they did not conclude anything on this. I think it is crucial to conclude here that resistance to fluoroquinolones in in non-pathogenic isolates M. hominis does not seem to be prevalent, at least this is what their data suggests.
- The authors claim that the study from Bebear et al. (ref [13] of the manuscript) found that gyrA mutations alone were able to induce ciprofloxacin resistance. However, this is not what is presented in this paper. All the resistant strains they found present GyrA and ParC mutations or GyrA and ParE mutations. I am also not sure why the authors refer ciprofloxacin resistance and not general Fluoroquinolone resistance, as the patterns of resistance found in the paper by Bebear et al are similar for all fluoroquinolones tested. I am not sure if this is a wrong citation or the authors need to explain what they mean by GyrA mutation only. This also leads me to the question of why the authors did not analyze the ParE mutations and specially the combination between ParE and GyrA mutations. The authors briefly say in the caption of figure 1 that no mutations were found in the QRD region of ParE. In any case, this should be included in the results section.
- It is not clear, at all, how the experiment of resistance development was performed. In the abstract the authors state that “both isolates spontaneously developed a second E84V in parC mutation and 26 resistance when sub-cultured in ciprofloxacin for 72-96 hours”. Later on the introduction the authors talk about resistance development after “ciprofloxacin or levofloxacin challenge”. However, in the material and methods section, the authors say “Induction of fluoroquinolone resistance was performed by allowing incubation of the MIC plates a further 24-72 hours beyond MIC determination (48 hours) and sub-culturing from sporadic colour changed wells above an MIC of 4 for levofloxacin.” The authors should clarify how these “resistance-development” experiments were carried out and which fluoroquinolones were used after all. It is especially important to explain why ciprofloxacin was used, if it was not tested for MIC.
- “The primary observation identified mutations of the QRDR region of parC corresponded to a fluoroquinolone resistant phenotype much more robustly than mutations in the QRDR of gyrA”. I do not understand what the authors mean with this first sentence of the discussion. Such an analysis is never presented in the results part. The only thing that is presented is that parC contained the highest levels of non-synonymous SNPS on the complete isolates tested. As only three of these isolates showed resistance against fluoroquinolones, could you explain what is the meaning of this sentence? Additionally, the structure of the discussion needs improvement. It lacks at least one sentence introducing the work in the problem of bacterial resistance.
Minor revisions:
The authors define the Quinolone Resistance Determining Region as QRDR and then systematically use QRDR regions. It should be changed to QRD regions to avoid repeating the word region.
Page 2 line 86 – examined and not examiend
Page 3 line 98 – it is sample U006 and not 006 (this error occurs frequently along the manuscript and should be corrected).
Page 6 line 115 – additionally is repeated
The authors should always use italic for M. hominis
Author Response
1. In the abstract the authors say: “However, there were high levels of polymorphism identified between all isolates outside of the QRDR, indicating caution for a genomics-led approach for resistance screening”. Why this never discussed in the manuscript?
The reviewer is absolutely correct. We did not capitalise on this observation or clarify it. In the discussion we have now added a comparison to the gyrA polymorphisms observed in a close relative Ureaplasma parvum in the discussion as well as citing related work. We have also related these observations to the very high diversity highlighted by ourselves and other authors in previous publications on developing MLST schemes for M. hominis.
2. On page 2, line 62, the authors claim that the mechanism of resistance to fluoroquinolones in M. hominis has been established as arising from SNPs in the QRD region of the topoisomerase genes but they do not present any reference for this. The only reference they present is a review that briefly explains this mechanism of resistance for E. coli. The authors should change this sentence. This mechanism of resistance was characterized for E.coli and the authors assume that it may be important in M. hominis as well, but it is not correct to say that this is the established mechanism of resistance in M. hominis, if no reference is presented. Alternatively, the authors should cite the work that presents this mechanism has the stablished one in M. Hominis.
The QRDR has been defined for many different bacterial species (gram positive, gram negative and cell wall-less), and out of convention the numbering is unified to the E.coli numbering system. This was further described in the discussion. However, we have added 2 references to further substantiate this statement for M.hominis. However, we have left the review for E.coli as well, as it is the most complete systematic review on fluoroquinolones available and their review is directly applicable to our findings.
3. Three isolates out of the 72 tested were resistant to levofloxacin and moxifloxacin, what about other fluoroquinolones, especially ciprofloxacin, were they tested? Why is ciprofloxacin later used to “trigger” resistance (according to the abstract) and it was not tested here? It would be important to have a table with the MICs (for all the fluoroquinolones tested) of the reference strains and, at least, these 3 isolates that showed resistance plus the 2 that carried the same gyrA mutation and developed resistance. I also suggest adding a supplementary table with the MICs for all isolates tested as it helps a lot following the results section.
We agree, the intermittent use of data pertaining to ciprofloxacin clouds the central issues, but is inherently problematic. Our study should have confined its details to those based on CLSI approved guidelines. Under those guidelines, there are no resistance thresholds identified for either ciprofloxacin or ofloxacin. As a result, only a very small subset of isolates were examined for ciprofloxacin, when replicating single step induction as outlined by Bebear et al. Our investigations of single step resistance were also performed in parallel with levofloxacin, with the same results; therefore, the references to ciprofloxacin have been removed. As only levofloxacin and moxifloxacin were used to determine MICs systematically across the whole cohort, an additional Figure has been added for these MICs as this was also requested by another reviewer.
4. In the discussion the authors say, correctly, that ciprofloxacin and ofloxacin are the fluoroquinolones more relevant in this study as they are the most frequently used. Then, they say that they found MICs for ciprofloxacin of 16-32 mg/L but do not state which isolates were resistant to this fluoroquinolone, if they had any mutation of the QRD region and do not show the data. This data is important and should be integrated and discussed in the paper. Again, it is not clear at all why ciprofloxacin was not used as a “test” compound but it was used in the “resistance development” experiments.
As detailed above, no internationally agreed resistance thresholds are available for any fluoroquinolones other than levofloxacin and moxifloxacin, therefore determination and reporting MICs are not meaningful. We have added more detail regarding this short-fall in the international guidelines for measuring resistance as well as indicating that ciprofloxacin MICs are routinely similar or higher than levofloxacin.
5. In the discussion the author acknowledge that they found a very low rate of resistance to fluoroquinolones in the set of samples they studied, when compared to other studies clinical isolates and that this is remarkable but they did not conclude anything on this. I think it is crucial to conclude here that resistance to fluoroquinolones in in non-pathogenic isolates M. hominis does not seem to be prevalent, at least this is what their data suggests.
These M. hominis isolates are not from non-pathogenic strains. All but 4 of the Cuban strains were from invasive infections, many of the Welsh clinical isolates were associated with urethritis and cervicitis as they were collected from patients attending sexual health clinic, as were the Serbian isolates. Therefore, we cannot make the requested amendment because it is not true. We thank the reviewer for pointing out that discrepancy, as on reflection the low levels of fluoroquinolone not completely unusual and we have extended the discussion to include recent studies with no to low fluoroquinolone resistance leves for cohorts of 85-100 strains. Therefore, we have amended the statement that our fluoroquinolone resistance prevalence was unusually low, which was not correct for European studies: UK, France and Serbia in 2019 (2/85 isolates; doi: 10.1093/jac/dkab320) and 0/100 M. hominis infection in patients attending a sexual health walk-in clinic in Wales (out of 1000 patients investigated in all; doi:10.1007/s10096-020-03993-7)
We account for the high levels of reported fluoroquinolone resistance to reliance on poor commercial assays (without appropriate CLSI-compliant follow-up of alleged resistance) that utilise inappropriately low concentrations of antibiotics such as ofloxacin and ciprofloxacin, for which no internationally agreed thresholds for resistance have been agreed. But this is outside of the scope of this manuscript.
6. The authors claim that the study from Bebear et al. (ref [13] of the manuscript) found that gyrA mutations alone were able to induce ciprofloxacin resistance. However, this is not what is presented in this paper. All the resistant strains they found present GyrA and ParC mutations or GyrA and ParE mutations. I am also not sure why the authors refer ciprofloxacin resistance and not general Fluoroquinolone resistance, as the patterns of resistance found in the paper by Bebear et al are similar for all fluoroquinolones tested. I am not sure if this is a wrong citation or the authors need to explain what they mean by GyrA mutation only. This also leads me to the question of why the authors did not analyze the ParE mutations and specially the combination between ParE and GyrA mutations. The authors briefly say in the caption of figure 1 that no mutations were found in the QRD region of ParE. In any case, this should be included in the results section.
The reviewer is absolutely correct. Upon re-reading reference 13, Bebear et al., also used norfloxacin, pefloxacin, and ofloxacin in addition to ciprofloxacin for multi-step induction of fluoroquinolone resistance. Further critical appraisal of that study found it was not that they didn’t find any parC mutations, but that they did not look at the parC or parE sequences at all. We have amended that part of the discussion to more accurately reflect these facts. However, it is important to note that our study examined the entirety of all gyrase and topoisomerase genes (including ParE) by whole genome sequence analysis. We found no SNPs that were found in ParE for any of the resistant isolates that did not also exist in susceptible strains as well. Furthermore, for those strains with GyrA mutations that were induced to resistance in a single step induction, no alteration between pre-resistant and post-resistant WGS was found for any part of the parE.
7. It is not clear, at all, how the experiment of resistance development was performed. In the abstract the authors state that “both isolates spontaneously developed a second E84V in parC mutation and 26 resistance when sub-cultured in ciprofloxacin for 72-96 hours”. Later on the introduction the authors talk about resistance development after “ciprofloxacin or levofloxacin challenge”. However, in the material and methods section, the authors say “Induction of fluoroquinolone resistance was performed by allowing incubation of the MIC plates a further 24-72 hours beyond MIC determination (48 hours) and sub-culturing from sporadic colour changed wells above an MIC of 4 for levofloxacin.” The authors should clarify how these “resistance-development” experiments were carried out and which fluoroquinolones were used after all. It is especially important to explain why ciprofloxacin was used, if it was not tested for MIC.
We apologise for the difficulty in conveying the single step induction of resistance method that we utilised. We have amended both results and methods text to make it more easily accessible to the reader. The reviewer also makes a good point regarding ciprofloxacin, we did not systematically investigate ciprofloxacin MIC and it was always used in parallel with levofloxacin for the small cohort of isolates that were examined for induction of resistance. It was included only because of the 1998 publication for induced resistance for this small cohort. Therefore, we have removed all reference to our use of ciprofloxacin in the manuscript. As there are no agreed international resistance thresholds for M. hominis and ciprofloxacin, it’s inclusion is confusing for the reader and cannot by justified due to the failure for CLSI guidelines to recognise it’s utility with this organism.
8. “The primary observation identified mutations of the QRDR region of parC corresponded to a fluoroquinolone resistant phenotype much more robustly than mutations in the QRDR of gyrA”. I do not understand what the authors mean with this first sentence of the discussion. Such an analysis is never presented in the results part. The only thing that is presented is that parC contained the highest levels of non-synonymous SNPS on the complete isolates tested. As only three of these isolates showed resistance against fluoroquinolones, could you explain what is the meaning of this sentence? Additionally, the structure of the discussion needs improvement. It lacks at least one sentence introducing the work in the problem of bacterial resistance.
We have clarified the first sentence of the discussion, as it appears the overall findings of the manuscript were not clearly stated. A mutation in the QRDR of GyrA is not sufficient to mediate fluoroquinolone resistance by the internationally agreed criteria as outlined by the CLSI, unless it is also accompanied by a mutation in ParC. While a S83L or E87G mutation in GyrA gives a slightly elevated MIC only for moxifloxacin.
Minor revisions:
The authors define the Quinolone Resistance Determining Region as QRDR and then systematically use QRDR regions. It should be changed to QRD regions to avoid repeating the word region.
We thank the reviewer for pointing this inadvertent replication out – we have unified it to QRDR and removed the word region, as QRDR is the more common convention over QRD region.
Page 2 line 86 – examined and not examined
This has been corrected.
Page 3 line 98 – it is sample U006 and not 006 (this error occurs frequently along the manuscript and should be corrected).
The use of U006 has been unified throughout the manuscript.
Page 6 line 115 – additionally is repeated
This has been corrected.
The authors should always use italic for M. hominis
This has been corrected.
Reviewer 2 Report
The work deals about a highly relevant theme, fluoroquinolone resistance in a variety of isolates, with different geographical origin. The obtained results could help in future studies, in order to better clarify potential functional effcts of the observed mutations. Surveillance strategies could have benefit from methods as those described in the present work.
Author Response
We thank the reviewer for their observations and concurrence.
Reviewer 3 Report
Tetracycline is the treatment of choice for Mycoplasma hominis infection. The tetracycline resistance M. hominis are treated with fluoroquinolones. The current manuscript is an extension of previous work the authors have investigated the susceptibility to fluoroquinolones in over 70 isolates of M. hominis and using whole-genome sequencing, they have provided the phylogenic variants and fluoroquinolones resistance gene signatures.
As compared to previous work they found around 4% resistance to fluoroquinolones. I find this work can benefit from a higher sample size. The manuscript can also use a clear explanation of the observed mutations and their probable mechanism, this could be added as a column in the appendix.
Please also provide the susceptibility data as a figure.
I could find only 1 figure in the current manuscript the authors have mentioned Figure 2 to the phylogenic tree. The current figure is missing figure legend, and the bottom part of the figure is missing. Apart from the need for more genomics surveillance author should also provide the significance of this work.
Author Response
Reveiwer: As compared to previous work they found around 4% resistance to fluoroquinolones. I find this work can benefit from a higher sample size. The manuscript can also use a clear explanation of the observed mutations and their probable mechanism, this could be added as a column in the appendix.
Response: We concur with the reviewer; however, these studies are limited by the availability of fluoroquinolone resistant isolates (we examined every isolate that we have collected since 2003) and by the extreme difficulty of scaling up sufficient M. hominis with high quality gDNA sufficient for WGS, and the extremely high cost for sequencing each isolate. It is worth noting that no previous publication has examined so many M. hominis isolates by WGS analysis. We have cited the best structural study on a related bacteria for QRDR mutations and their probably mechanism (reference doi:10.1128/AAC.02560-19) in the discussion. The QRDR mutations are also highlighted in red for Appendix Tables A1 and A3.
Reviewer: Please also provide the susceptibility data as a figure.
Response: We have added a new figure as requested.
Reviewer: I could find only 1 figure in the current manuscript the authors have mentioned Figure 2 to the phylogenic tree. The current figure is missing figure legend, and the bottom part of the figure is missing. Apart from the need for more genomics surveillance author should also provide the significance of this work.
Response: I am not sure how this happened. We will ensure that the PDF generated for the revised version does not displace or shift the figures.